# A Theranostic Approach in SIRT: Value of Pre-Therapy Imaging in Treatment Planning

**DOI:** 10.3390/jcm11237245

**Published:** 2022-12-06

**Authors:** Philippe d’Abadie, Stephan Walrand, Renaud Lhommel, Michel Hesse, François Jamar

**Affiliations:** Department of Nuclear Medicine, Cliniques Universitaires Saint Luc, Université Catholique de Louvain, 1200 Brussels, Belgium

**Keywords:** theranostic, SIRT, radioembolization, pre-therapy imaging, MAA SPECT/CT

## Abstract

Selective internal radiation therapy (SIRT) is one of the treatment options for liver tumors. Microspheres labelled with a therapeutic radionuclide (^90^Y or ^166^Ho) are injected into the liver artery feeding the tumor(s), usually achieving a high tumor absorbed dose and a high tumor control rate. This treatment adopts a theranostic approach with a mandatory simulation phase, using a surrogate to radioactive microspheres (^99m^Tc-macroaggregated albumin, MAA) or a scout dose of ^166^Ho microspheres, imaged by SPECT/CT. This pre-therapy imaging aims to evaluate the tumor targeting and detect potential contraindications to SIRT, i.e., digestive extrahepatic uptake or excessive lung shunt. Moreover, the absorbed doses to the tumor(s) and the healthy liver can be estimated and used for planning the therapeutic activity for SIRT optimization. The aim of this review is to evaluate the accuracy of this theranostic approach using pre-therapy imaging for simulating the biodistribution of the microspheres. This review synthesizes the recent publications demonstrating the advantages and limitations of pre-therapy imaging in SIRT, particularly for activity planning.

## 1. Introduction

Selective internal radiation therapy (SIRT) is a treatment option for liver tumors by delivering radioactive microspheres in the tumors’ feeding arteries, leading to very high absorbed doses to tumors and usually achieving a high tumor control rate [1]. Hepatic tumors are mainly vascularized by the liver artery as opposed to the healthy liver, whose blood supply is preferentially ensured by the portal vein [2]. This unique characteristic is the milestone of SIRT efficacy. By injecting radioactive microspheres in the liver artery or any of its branches, tumor(s) receive very high absorbed doses while sparing the healthy liver.

Three types of radioactive microspheres are commercially available, differing by their physical and radioactive characteristics: yttrium-90 (^90^Y)-resin microspheres (Sir-Spheres^®^, Sirtex Medical GmbH, Sydney, Australia), ^90^Y-glass microspheres (Therasphere^®^, Boston Scientific, Boston, MA, USA), and holmium-166 (^166^Ho)-poly-L-lactic acid (PLLA) microspheres (QuiremSpheres^®^, Quirem Medical B.V., Deventer, The Netherlands) [3]. Only Sir-Spheres^®^ and Therasphere^®^ have FDA approval for the U.S. market.

The SIRT procedure applies a theranostic approach by using similar radiopharmaceuticals agents for both therapy planning and treatment [4]. During the first step, the treatment is simulated using a surrogate: ^99m^Tc-macroaggregated albumin (MAA) or a scout dose of the radioactive microspheres (^166^Ho-scout dose), enabling a selection of patients who have good tumor targeting and no absolute contraindications for treatment such as excessive lung shunt and digestive uptake. These radionuclides emit low energy gamma rays (Table 1), allowing for detection by nuclear imaging systems to precisely localize the distribution in the liver or in non-targeted tissues. In particular, 3D imaging using single photon emission computed tomography combined with CT scan (SPECT/CT) allows an accurate evaluation of the MAA distribution (MAA SPECT/CT) for evaluating extra hepatic arterial shunting to limit the risks of lung and gastrointestinal complications during treatment [5,6].

The MAA distribution in the tumor and healthy liver compartments can also be analyzed to estimate the absorbed doses (multicompartmental model) and establish the amount of radioactive microspheres to be scheduled for the treatment (activity planning) [7].

Using ^166^Ho microspheres, a minimal dose (scout dose) can be used as an alternative to MAA particles. This scout dose can simulate the treatment without risks of radiation toxicity [8].

After treatment, the correct distribution of the radioactive microspheres is confirmed by SPECT/CT for ^90^Y (bremsstrahlung) and for ^166^Ho by ^90^Y PET/CT (radiation characteristics in Table 1). In addition, the paramagnetic nature of holmium metal enables MRI imaging, evaluating as well the biodistribution of Ho microspheres [9].

Post-therapy ^90^Y PET/CT imaging evaluates with accuracy the actual absorbed doses to the tumor(s) and to the healthy liver [3].

Pre-therapy MAA SPECT/CT/CT and post-therapy ^90^Y PET/CT absorbed doses are strongly correlated with tumor response and patient outcome (high tumor absorbed dose) or with liver toxicity (high healthy liver absorbed dose) [10,11,12,13].

Figure 1 illustrates pre-therapy and post-therapy imaging in a patient treated by resin microspheres for colorectal metastases, showing a nice match between ^99m^Tc-MAA and ^90^Y-resin spheres distribution.

With regards to activity planning, different simple and safe methods are recommended by manufacturers. Activity planning is performed using a monocompartmental model, targeting a liver absorbed dose of 80–150 Gy with glass microspheres and 60 Gy for ^166^Ho microspheres [3]. For resin microspheres, activity planning is often performed with a semi-empirical method, using a calculation based on the patient’s body surface area and the tumor burden. As an alternative, Sirtex Medical also recommends the use of a more complex method named the partition model or multicompartmental model, which takes into account separately the absorbed doses to the non-tumoral liver, to the tumor(s), and to the lungs using MAA pre-therapy imaging [14]. Article 56 of the EU Council Directive 2013/59 and recent guidelines recommend the use of this personalized method of activity planning, differentiating tumor and healthy liver compartments [15,16,17]. As demonstrated by previous data, this multicompartmental model optimizes treatment planning and improves the effectiveness of SIRT [13]. With this model, the DOSISPHERE-01 randomized controlled trial demonstrated a significant increase in tumor response and better patient outcome in hepatocellular carcinoma (HCC), without increasing the toxicity [10].

This review aims to evaluate the accuracy of pre-therapy imaging using MAA or a scout dose for predicting the treatment efficacy. This review will synthesize the recent publications demonstrating the advantages and limitations of pre-therapy imaging in SIRT, particularly for activity planning.

## 2. MAA Particles as a Surrogate to Radioactive Microspheres: Physical and Technical Limitations

Prior to SIRT, a mandatory simulation phase is performed with diagnostic liver arteriography for mapping the tumor vascularization coil embolization if needed (e.g., in case of proximity to arterial digestive branches) and ends with the injection of MAA particles through the angiographic catheter, well positioned in the vascular territory of the tumor(s). This angiographic procedure is followed by planar and SPECT/CT imaging of the MAA distribution. Like ^90^Y or ^166^Ho microspheres, MAA is trapped in the arterial microvasculature of the liver and can therefore simulate the distribution of the microspheres. Nevertheless, MAA is not the perfect surrogate because of their physical characteristics, which differ from those of the therapeutic microspheres.

Compared to microspheres, MAA particles have a variable shape, a different size distribution (90% within 10 to 90 μm) with a lower mean size (15 μm versus 25–32 μm), and are injected in much smaller numbers (Table 2) [3,18,19,20]. These differences can explain some discrepancy between the MAA and the microsphere biodistribution, especially the risk of increased shunt to the lungs with MAA.

The smallest MAA particles can pass through the liver capillaries and be responsible for over-estimation of the lung shunt [21]. Nevertheless, an excessive MAA uptake in the lungs indicates high hepato-pulmonary shunting and hence a relative contraindication for SIRT (risks of radiation-induced pneumonitis and fibrosis) [22]. A lung shunt superior to 20% or an estimated lung absorbed dose in excess of 30 Gy are contraindications for SIRT [23]. Moreover, these physical differences can be responsible for distribution variations in the healthy and tumor(s) compartments. More precisely, the variable size of MAA particles seems to be a limiting factor in tumor absorbed dose prediction. Indeed, by simulating the treatment with a scout dose of microspheres instead of MAA particles, tumor absorbed doses were more precisely predicted [24,25].

In addition, the lack of precise replication of positioning between the liver arteriographies at the time of simulation and treatment can result in a mismatch between liver distribution of MAA and microspheres. Both procedures require the use of similar angiographic catheters and positioning as well as a well-matched orientation in the artery lumen, and end with an injection of particles at a distance from arterial bifurcations, after controlling for the absence of any vasospasm [26,27]. In addition, MAA particles must also be injected slowly (over 20 s) to better match with the treatment procedure [28].

## 3. Accuracy of MAA SPECT/CT for Predicting Tumor and Non-Tumoral Whole Liver Absorbed Doses

Some previous studies compared the simulated MAA tumor and non-tumoral liver absorbed doses to the real absorbed doses calculated with post-therapy ^90^Y imaging. The non-tumoral whole liver absorbed dose (NTWLD) refers to the whole healthy liver (targeted or not by SIRT) as defined recently by the EANM dosimetry committee [18]. In these studies, the strength of the correlation between the MAA absorbed doses and the ^90^Y absorbed doses was assessed and quantified using the Pearson coefficient R, ranging from 0 (no linear correlation) to 1 (perfect linear correlation) [29]. Figure 2 represents the results of previous studies where the Pearson test was used for comparing the biodistribution of MAA and the radioactive microspheres (more details in Appendix A). Regarding the estimate of tumor absorbed doses, they demonstrated a variable correlation, with a Pearson coefficient ranging from 0.56 to 0.91. Nevertheless, a large majority of these studies demonstrated a moderate correlation (R < 0.7), and only two studies with a small number of patients showed a higher correlation (R close to 0.9). Another parameter was also compared, i.e., tumor to liver ratio (TNR), and will be discussed further.

Some studies also measured the agreement between these absorbed doses using Bland–Altman analyses [41]. By measuring the differences between these pairs, this statistical test evaluated the risk of errors in MAA imaging for calculating the absorbed doses, after defining the limits of agreement (95% confidence interval). Table 3 summarizes studies performing Bland–Altman analyses and where the relative confidence intervals were available or estimated from absolute values. All these studies demonstrated a similar risk of under- and over-estimation of the tumor absorbed dose (similar plotting over and under the reference X axis). Moreover, the limits of agreement were large in a majority of these studies, with a maximum error over 100% of the tumor absorbed dose estimation in 3 studies.

Regarding the prediction of the normal liver absorbed dose, all studies were congruent and demonstrated the accuracy of MAA SPECT/CT. The NTWLD calculated with pre-therapy MAA SPECT/CT was strongly correlated with the NTWLD calculated with post-therapy ^90^Y PET/CT, with a Pearson coefficient over 0.9 in all studies (Figure 2, Appendix A). Moreover, Bland–Altman analyses confirmed this good agreement with a maximal deviation ranging from −36% to +36% (Table 3).

Using alternative methods of comparison in 16 patients, Thomas et al. showed similar results, demonstrating a lack of prediction of the tumor absorbed dose but a highly reliable prediction of the normal liver absorbed dose with MAA imaging [43].

## 4. Value of MAA SPECT/CT to Predict the Tumor to Normal Liver Uptake Ratio

Some recent studies compared pre-therapy MAA imaging to post-therapy ^90^Y imaging using the tumor to normal liver uptake ratio (TNR). As previously mentioned, tumors are preferentially vascularized by the liver artery while the healthy liver is vascularized by the portal vein; therefore, TNR is usually greater than 1 [44]. This ratio is also directly correlated to the tumor absorbed dose [45]. The results of these studies are summarized in Figure 2B and detailed in Appendix A. They demonstrated results similar to the prediction of tumor absorbed doses. Indeed, TNR was moderately predicted, with a Pearson coefficient ranging from 0.53 to 0.9.

Other studies were performed using an alternative Spearman’s rank test, also demonstrating a moderate correlation in a study analyzing colorectal metastases (ρ = 0.51) and in a study pooling a mix of tumors (ρ = 0.65) [45,46].

Nevertheless, a high TNR uptake seems very well predicted by MAA imaging. In a study evaluating 171 tumors, MAA SPECT/CT predicted a high TNR ratio (≥1.5) with an accuracy of 85% [34]. Moreover, in liver metastases from colorectal cancer (CRC), a TNR ratio superior to 1.7 was a predictor of SIRT efficacy [45]. Therefore, the TNR ratio estimated from MAA imaging can generally select patients suitable for SIRT, avoiding ineffective procedures for patients with TNR ratios close to 1 [13,17].

## 5. MAA Tumor Absorbed Doses Correlate with Clinical Outcome after SIRT

Despite the low accuracy of MAA SPECT/CT for predicting tumor absorbed doses, numerous data are available for demonstrating a strong relationship between the MAA tumor absorbed dose and SIRT efficacy. In particular, when a certain MAA tumor absorbed dose threshold was achieved, the radiological response and the patient outcome were significantly improved. Table 4 summarizes studies showing this correlation between MAA tumor absorbed doses and the clinical outcome, demonstrating the high significance of this pre-therapy dosimetry. Nevertheless, a distinction must be made between determining an exact tumor absorbed dose and predicting a high level of tumor absorbed dose. Based on previous data, an exact tumor absorbed dose cannot be predicted from MAA imaging, but its level (low versus high) can be determined with high accuracy. When a high tumor absorbed dose is simulated with MAA SPECT/CT, the tumor will receive a high tumor absorbed dose in a large majority of the cases, achieving a threshold of high tumor absorbed dose. In a series of patients treated for HCC by resin microspheres, a threshold dose of 100 Gy simulated with MAA imaging was achieved in 90% of them on ^90^Y post-therapy imaging [34].

## 6. Dosimetry Considerations Using MAA SPECT/CT

Using the multicompartmental model with MAA SPECT/CT, activity planning can be determined to reach a tumor absorbed dose threshold. Regarding HCC, it is recommended to target a tumor absorbed dose equal to or above 100–120 Gy with resin microspheres and equal to or above 205–300 Gy with glass microspheres [16,17].

Two dosimetric factors predict SIRT efficacy: a high tumor absorbed dose and homogeneous distribution of the microspheres in the tumor [3,11,56]. The tumor response rate will proportionally increase with the absorbed dose, reaching a plateau of complete response for the highest doses [50,55,56,57,58]. From a dosimetric point of view, the main limitation of SIRT effectiveness is the heterogeneity of the microsphere distribution in the tumor [3]. Tumor absorbed doses can be corrected for heterogeneity using Equivalent Uniform Dose (EUD) [59]. In a series of HCC tumors, the tumor control probability reached 95% for an EUD over 100 Gy, estimated with post therapy ^90^Y PET/CT [56]. This EUD was achieved for a large range of tumor absorbed doses, varying from 190 (homogeneous distribution) to 1800 Gy (heterogeneous distribution). Therefore, targeting a precise tumor absorbed dose with MAA SPECT/CT seems inappropriate, and the tumor absorbed dose must be the highest possible for increasing the probability of complete response. In addition, as demonstrated in previous data (Table 3), the tumor absorbed dose is overestimated in approximately half of cases with MAA imaging, resulting sometimes in very low and ineffective tumor absorbed doses, as demonstrated in Table 5. Table 5 estimates the confidence intervals of the real (post-therapy) tumor absorbed dose by targeting a precise MAA tumor absorbed dose (derived from data in Table 3). Therefore, the recommended thresholds of tumor absorbed doses targeted with MAA imaging must be considered with caution, and one should always aim for the highest tumor absorbed dose taking into account the liver tolerance. The DOSISPHERE-01 trial illustrated well this concept by adopting an interesting method of activity planning in the arm of patients treated with the multicompartmental model, trying to reach very high levels of tumor absorbed doses, beyond the recommended thresholds [10]. Indeed, the activity was planned to target a tumor absorbed dose over 205 Gy but, if possible, over 250 Gy. In this study arm, the mean tumor absorbed dose planned with MAA SPECT/CT ended up at 332 Gy, largely above the recommended thresholds. Thereafter, the clinical efficacy of this method was strongly confirmed during the follow up.

Moreover, due to the low predictive value of MAA imaging for tumor absorbed doses, the thresholds of MAA tumor absorbed doses cannot be used for patient selection during the workup. Indeed, a tumor absorbed dose threshold not reached with pre-therapy MAA dosimetry can be finally reached with post-therapy dosimetry. This was demonstrated by previous data analyzing post-therapy ^90^Y tumor dosimetry, showing a clinical efficacy under these MAA recommended tumor absorbed dose thresholds [13].

Using the multicompartmental model, an alternative strategy can be used: targeting the maximum tolerable absorbed dose to the non-tumoral liver [12,34]. This method can be used in patients with a TNR significantly higher than 1. Therefore, the activity can be significantly increased, as can the tumor absorbed dose and then the tumor control probability [34].

Activity planning is limited by the risk of toxicity related to the radiation of the healthy liver. Radioembolization-induced liver disease (REILD) is the more serious complication of SIRT, occurring in less than 5% of patients. REILD is defined by liver damage occurring within six months after SIRT in absence of tumor progression [16]. Patients with underlying liver disease and especially advanced cirrhosis (elevated baseline bilirubinemia or Child score B) and a low liver reserve (<30% of liver not targeted by SIRT) are at higher risk [60,61]. Additionally, REILD is strongly correlated with the NTWLD, with a significant risk for absorbed doses above 40–50 Gy with resin microspheres and above 70–90 Gy with glass microspheres [12,57,62,63].

In addition, using glass microspheres, Garin et al. demonstrated permanent liver toxicity (≥grade 3) for patients with a healthy liver dose in the targeting liver above 120 Gy when the liver reserve was inferior to 30% [51,61]. Interestingly, 120 Gy in 70% of the healthy liver corresponds to a NTWLD of 80 Gy, equal to the well-known threshold of toxicity. Radiobiological models demonstrated that liver tolerance to radiation is dependent on the liver volume and is very strong for an irradiated healthy liver volume inferior to 40%. In addition, the absorbed dose averaged from the whole normal liver was the strongest parameter correlated with liver toxicity [18]. Therefore, by injecting an activity reaching or staying behind a limit of NTWLD of 40 Gy for resin microspheres and 70 Gy for glass microspheres, the treatment planning can be significantly optimized for clinical efficacy while controlling for the risk of liver toxicity. Indeed, the accurate prediction of the healthy liver dose with MAA imaging allows for performing this safe planning. In the worst scenario, the NTWLD will stay around the safe thresholds (Table 5).

The formula for calculating the activity based on this non-tumoral liver absorbed dose reads as follows [34]:A=CWLCNTL. NTLD.  MNTL. (1+LSF)50

With *A*: the activity planned (GBq), *C_WL_*: the counts in the whole liver defined on MAA SPECT, *C_NTL_*: the counts in the non-tumoral liver defined on MAA SPECT, *NTLD*: the established absorbed dose to the non-tumoral liver (Gy), *M_NTL_*: the mass of the non-tumoral liver (kg), *LSF*: the lung shunt fraction.

## 7. Pre-Therapy Imaging with ^166^Ho SPECT/CT

^166^Ho (PLLA) microspheres can be used as an alternative to ^90^Y (resin or glass) microspheres. The ^166^Ho radionuclide emits a beta radiation responsible for its therapeutic effect but also a small amount of gamma radiation (Table 1) that allows imaging by a SPECT system (^166^Ho SPECT/CT) [3]. ^166^Ho scout dose microspheres (maximal activity of 250 MBq) is a safe alternative to MAA particles for evaluating the feasibility of the treatment [8].

^166^Ho scout microspheres have the same shape and size as ^166^Ho microspheres used for therapy. Therefore, pre-therapy ^166^Ho SPECT/CT is expected to be more predictive than MAA SPECT/CT. A previous study comparing these imaging methods demonstrated that the lung absorbed dose was more precisely evaluated with this scout dose compared to MAA imaging [64]. Indeed, MAA particles can overestimate the lung shunt due to a fraction of some small particles (i.e., 10 μm) which can pass through the liver capillaries and reach the lungs [65]. Regarding the intrahepatic biodistribution, pre-therapy ^166^Ho SPECT/CT can also estimate the healthy liver and tumor absorbed doses. Smits et al. demonstrated similar values of pre-therapy ^166^Ho SPECT/CT and pre-therapy MAA SPECT/CT for predicting the healthy liver absorbed dose. Regarding the tumor absorbed dose, pre-therapy ^166^Ho SPECT/CT was more predictive than MAA SPECT/CT, but the absolute confidence intervals of Bland–Altman analyses were still significant, demonstrating only a moderate prediction (CI 95%: −90.3 Gy; + 105.3 Gy) [24].

The predictive value of the ^166^Ho scout dose is also limited by the spatial resolution of ^166^Ho SPECT/CT. Indeed, the ^166^Ho radionuclide emits a multitude of gamma radiations of high energy (MeV range), which interact with the patient’s body and mainly with the lead collimator of the SPECT/CT [66]. Therefore, secondary lead X-rays are produced that fall in the energy window set around the 81 keV photopeak of ^166^Ho, limiting significantly the spatial resolution and the quantitative assessment of the microsphere distribution. Last generation SPECT/CT systems using tungsten collimators would not be affected by this problem, improving in theory the spatial resolution and the quantification. Additionally, a loss of the spatial resolution is also due to the use of a medium energy collimator used during the SPECT acquisition [67]. Therefore, previous dosimetry studies used complex reconstruction algorithms, with Monte Carlo simulation modelling, for improving the resolution of the SPECT/CT and the accuracy of dosimetry [8,21,24,66,68,69]. These Monte Carlo simulations are for now only used in research and development, involving important computer processing resources and time, with software that is not commercialized nor available for routine clinical use. Therefore, the monocompartmental model is recommended using ^166^Ho microspheres, targeting a maximum of 60 Gy to the whole liver volume [70]. Nevertheless, a recent study evaluating post-therapy ^166^Ho SPECT/CT dosimetry demonstrated a dose response relationship and a better outcome for patients treated with a tumor absorbed dose of at least 90 Gy (for colorectal metastases). In addition, no liver toxicity was shown in this population of patients with a NTWLD reaching 55 Gy [69]. Similar studies evaluating dose effect relationships using the pre-therapy ^166^Ho scout dose would support in the future the use of the partition model for activity planning.

## 8. Conclusions

Pre-therapy imaging in SIRT is strongly involved in a theranostic approach for conducting treatment planning. MAA SPECT/CT plays a crucial role in selecting potential good responders to SIRT. Patients with high MAA tumor uptake have a high probability of receiving high and efficient tumor absorbed doses after SIRT. Moreover, pre-therapy imaging accurately predicts the healthy liver absorbed dose and, therefore, the planned therapeutic activity can be optimized while still staying behind the threshold dose of liver toxicity. Using this model, treatment planning really becomes personalized, improving the safety and the clinical benefits of SIRT.

## Figures and Tables

**Figure 1 jcm-11-07245-f001:**
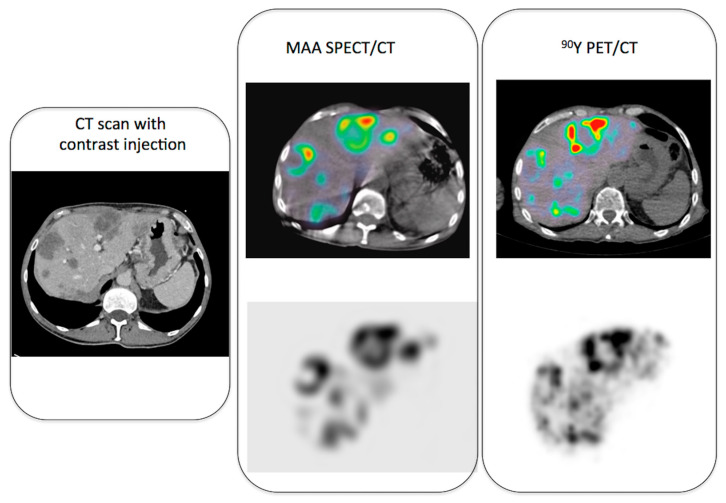
Example of pre-therapy imaging (MAA SPECT/CT) and post-therapy imaging (^90^Y PET/CT) in a patient with colorectal metastases.

**Figure 2 jcm-11-07245-f002:**
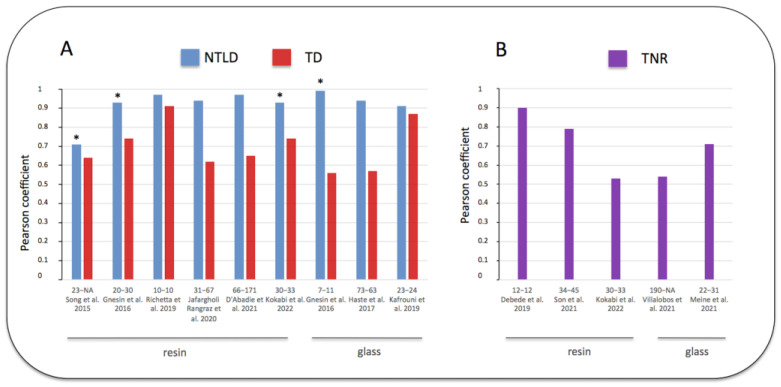
Pearson coefficient correlations between doses estimated by MAA SPECT/CT and post ^90^Y SIRT imaging [25,30,31,32,33,34,35,36,37,38,39,40]. (**A**) blue: non-tumoral liver absorbed dose (NTLD), referring to the non-tumoral whole liver or to the non-tumoral liver targeted by SIRT (*). (**A**) red: tumor absorbed dose (TD). (**B**) purple: tumor to normal liver ratio uptake (TNR). Abscissa legend: patients and tumor numbers of each study referenced in bracket. Appendix A provide additional information. Note the better correlation obtained for the NTLD versus TD or TNR.

**Table 1 jcm-11-07245-t001:** Radiation emission characteristics of radiolabeled microspheres and MAA detected by nuclear imaging systems.

	Labelled Particle	Gamma	Positron
Radionuclide		Emission Detected by SPECT/CT	Emission Detected by PET/CT
(Radioactive Half Life)		(Energy and Abundance)	(Abundance)
^99m^Tc	MAA	Direct g radiation (140 keV, 89%)	NA
(6 h)			
^166^Ho	PLLA microspheres	Direct g radiation (82 KeV, 6,7%)	NA
(27 h)			
^90^Y	resin and glass microspheres	Indirect radiation (bremsstrahlung, continuous spectrum radiation from 0 to 2.3 MeV)	0.0036%
(64 h)			

NA: not available.

**Table 2 jcm-11-07245-t002:** Main physical characteristics of MAA and radioactive microspheres.

Characteristics	^99m^Tc-MAA Particles	^90^Y-Resin Microspheres	^90^Y-Glass Microspheres	^166^Ho-PLLA Microspheres
Diameter (mean)	15 μm	32 μm	25 μm	30 μm
Usual number of injected particles (in millions)	0.3–0.7	50 *	4 *	30 *

* For a 3 GBq activity at usual calibration time.

**Table 3 jcm-11-07245-t003:** Predictive value of pre-therapy MAA imaging for estimating TD and NTWLD (using Bland–Altman analyses for measuring the extremes).

Studies	Type of Microspheres	Post Therapy Imaging	Type of Tumors	Nb of Patients	Nb of Tumors	TD Estimation (Relative 95% CI) ^+^	NTWLD Estimation (Relative 95% CI) ^+^
Kafrouni et al. 2019 [36]	glass	^90^Y PET/CT	HCC	23	24	−29%; +29%	−34%; +34%
Richetta et al. 2019 [32]	resin	^90^Y PET/CT	HCC	10	10	−39%; +33%	−24%; +31%
Jafargholi Rangraz et al. 2020 [33]	resin	^90^Y PET/CT	HCC and mets	31	67	−169%; +146%	−30%, +23%
Jadoul et al. 2020 [42]	resin and glass	^90^Y PET/CT	HCC and mets	57	137	−100%; +100%	−36%, +36%
d’Abadie et al. 2021 [34]	resin	^90^Y PET/CT	HCC and mets	66	171	−76%; +320%	−19%, +24%

HCC: hepatocellular carcinoma; mets: metastases; TD: Tumor absorbed dose; Nb: number; NTWLD: non-tumoral whole liver absorbed dose; ^+^ In some studies, absolute 95% CI were converted in relative 95% CI (comparison to the mean).

**Table 4 jcm-11-07245-t004:** MAA tumor absorbed doses thresholds correlated to SIRT efficacy.

Studies ^+^	Microspheres	Tumors	Number of Patients	Tumor Absorbed Dose Threshold	Predictor of Tumor Response	Predictor of Better Survival (Median)
Chiesa et al. 2011 [47]	glass	HCC	46	≥257 Gy	85%	NA
Garin et al. 2012 [48]	glass	HCC	36	≥205 Gy	100%	18 mo ^°^
						(vs. 9 mo)
Mazzaferro et al. 2013 [49]	glass	HCC	52	≥500 Gy	80%	NA
Chiesa et al. 2015 [50]	glass	HCC	52	≥217 Gy	100%	NA
Garin et al. 2017 [51]	glass	HCC	85	≥205 Gy	98%	21 mo ^°^
						(vs. 6.5 mo)
Bourien et al. 2019 [52]	glass	CGC	64	≥260 Gy	88% *	28.2 mo ^°^
						(vs. 11.4 mo)
Lam et al. 2013 [53]	resin	CRC	25	≥55 Gy	NA	32.8 mo ^°^
						(vs. 7.2 mo)
Chansanti et al. 2017 [54]	resin	NET	15	≥191 Gy	83%	NA
Piasecki et al. 2018 [45]	resin	CRC	21	≥70 Gy	99%	NA
Hermann et al. 2020 [55]	resin	HCC	121	≥100 Gy	72%	14.1 mo ^°^
						(vs. 6.8 mo)
Son et al. 2021 [38]	resin	HCC	34	≥125 Gy	86%	NA

HCC: hepatocellular carcinoma; CGC: cholangiocarcinoma; NET: neuroendocrine tumors; CRC: colorectal cancer; NA: not available; * subgroup of 29 patients; ^+^ Studies with mixed tumor types and without distinction of types of treatment (resin or glass) were excluded. ^°^ differences statistically significant (*p* < 0.05).

**Table 5 jcm-11-07245-t005:** Confidence intervals of the real post-therapy absorbed doses after targeting doses on MAA imaging for HCC tumors. 95% confidence intervals are based on the results of previous studies shown in Table 3.

	TD		NTWLD		Studies
Pre-Therapy (MAA) Target Dose	100 Gy (Resin)	205 Gy (Glass)	40 Gy (Resin)	70 Gy (Glass)	Nb Patients- Nb Tumors [Reference]
**Confidence intervals of the real absorbed dose** **[95% CI]**	[71–129 Gy]	[146–264 Gy]	[26–54 Gy]	[46–94 Gy]	23–24
					[36]
	[61–133 Gy]	[125–273 Gy]	[30–52 Gy]	[53–92 Gy]	10–10
					[32]
	[0–246 Gy]	[0–504 Gy]	[28–49 Gy]	[49–86 Gy]	31–67
					[33]
	[0–200 Gy]	[0–410 Gy]	[26–54 Gy]	[45–95 Gy]	57–137
					[42]
	[24–420 Gy]	[49–861 Gy]	[32–50 Gy]	[57–87 Gy]	66–171
					[34]

TD: Tumor absorbed dose; NTWLD: non-tumoral whole liver absorbed dose; CI: confidence intervals.

## Data Availability

No new data were created in this study. Data sharing is not applicable in this article.

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
