# Peer review of "A Theranostic Approach in SIRT: Value of Pre-Therapy Imaging in Treatment Planning"

_jcm, 2022, doi:10.3390/jcm11237245_

Round 1

Reviewer 1 Report

This manuscript allows an exhaustive review of the available evidence on the usefulness of MAA and of 166Ho scout as surrogate predictors of 90Y SIRT and 166Ho SIRT, respectively. However, for a better understanding, this reviewer considers some clarifications/modifications necessary:

1.      The term “whole normal liver absorbed dose” can be misleading. It must be clearly defined. As it appears, it seems to correspond to the term defined by Carlo Chiesa to refer to the entire healthy liver which is widely used in treatments with glass microspheres. However, some of the articles commented on in the text (ie, Gnesin S et al, J Nucl Med 2016), did not refer to the entire volume of healthy tissue, but to the healthy tissue that received SIRT (non tumoral volume). Please check which authors refer to one or another term.

2.      Page 10, line 333 : “Therefore, only the monocompartmental model is recommended using 166 Ho microspheres, targeting 60 Gy in maximum to the whole liver volume”, contradicts what was mentioned by Caren van Roeckel et al. J Nucl Med 2021 “Dose-Effect Relationships of 166Ho Radioembolization in Colorectal Cancer”: “For future patients, it is advocated to use a 166Ho scout dose to select patients and yo personalize the administered activity, targeting a mean tumor-absorbed dose of more than 90 Gy and a parenchymal dose of less than 55 Gy”.

3.      The statement in line 73 page 2 that “post-therapy imaging evaluates with accuracy the actual dose” is, at least, debatable for the bremsstrahlung imaging.

4.      Page 3 line 88: “With regards to activity planning, manufacturers recommend basic but safe dosimetry methods: a calculation based on the body surface area for resin microspheres..” it is not accurate. The SIR-sphere manufacturer also recommends the partition method in the IFU.

5.      Table 5 is difficult to understand, numbers in square brackets are not described in the table foot.

6.      Page 9, line 263: “Using the multicompartmental model, an alternative strategy can be used: targeting the maximum tolerable absorbed dose to the healthy liver independently of a tumor threshold dose”. It seems more accurate to make this recommendation once it is known that the tumor is going to receive at least the minimum tumoricidal dose for each type of microsphere.

7.      Other minor comments are:

-page 1, line 42: Sirtex Medical Ltd. has change to Sirtex Medical GmBH

-The use of more commonly used nomenclatures and therefore more easily recognizable is recommended, such as TN or TNR for Tumor/Lesion (instead of TL) or CGC for Cholangiocarcinoma (instead of CK)

-please, change pneumonia by pneumonitis (page 4, line 125)

Author Response

Dear reviewer,

We thank you very much for your review and your interesting comments.

Comments reviewer 1:

This manuscript allows an exhaustive review of the available evidence on the usefulness of MAA and of 166Ho scout as surrogate predictors of 90Y SIRT and 166Ho SIRT, respectively. However, for a better understanding, this reviewer considers some clarifications/modifications necessary:

  1. The term “whole normal liver absorbed dose” can be misleading. It must be clearly defined. As it appears, it seems to correspond to the term defined by Carlo Chiesa to refer to the entire healthy liver which is widely used in treatments with glass microspheres. However, some of the articles commented on in the text (ie, Gnesin S et al, J Nucl Med 2016), did not refer to the entire volume of healthy tissue, but to the healthy tissue that received SIRT (non tumoral volume). Please check which authors refer to one or another term.

Response: Indeed, the whole normal liver absorbed dose refers to the entire healthy liver (targeted or not by SIRT) and can be misunderstood. This term was changed by the non-tumoural whole liver absorbed dose (NTWLD), as defined recently by the EANM dosimetry committee (Chiesa et al. EJNMMI Physics 2021).

The non-tumoural whole liver absorbed dose was defined in chapter 3 as follows (P4 line 148): “The non-tumoural whole liver absorbed dose (NTWLD) refers to the whole healthy liver (targeted or not by SIRT), as defined recently by the EANM dosimetry committee [18]. »

Regarding the publication from Gnesin et al. 2016, it is indeed a mistake, for other authors, there is no change after verification. Figure 2 and table S1 were modified as required.

  1. Page 10, line 333 : “Therefore, only the monocompartmental model is recommended using 166 Ho microspheres, targeting 60 Gy in maximum to the whole liver volume”, contradicts what was mentioned by Caren van Roeckel et al. J Nucl Med 2021 “Dose-Effect Relationships of 166Ho Radioembolization in Colorectal Cancer”: “For future patients, it is advocated to use a 166Ho scout dose to select patients and to personalize the administered activity, targeting a mean tumor-absorbed dose of more than 90 Gy and a parenchymal dose of less than 55 Gy”.

Response: It is not a real contradiction in our opinion because dosimetry in the study of Van Roeckel et al. was performed with post therapy 166Ho SPECT/CT, not with the pre therapy Quirem scout dose. Nevertheless, you are true, it is promising for the future.

These sentences were added to the manuscript (P10 line 349):

« Nevertheless, a recent study evaluating post therapy 166 Ho SPECT/CT dosimetry demonstrated a dose response relationship and a better outcome for patients treated with a tumor absorbed dose of at least 90 Gy (for colorectal metastases). In addition, no liver toxicity was shown in this population of patients with a NTWLD reaching 55 Gy [70]. Similar studies evaluating dose effect relationships using the pre-therapy 166Ho scout dose would support in the future the use of the partition model for activity planning. »

  1. The statement in line 73 page 2 that “post-therapy imaging evaluates with accuracy the actual dose” is, at least, debatable for the bremsstrahlung imaging.

Response: Thank you for this remark, we changed by “Post-therapy 90Y PET/CT imaging” P2 line 75

  1. Page 3 line 88: “With regards to activity planning, manufacturers recommend basic but safe dosimetry methods: a calculation based on the body surface area for resin microspheres..” it is not accurate. The SIR-sphere manufacturer also recommends the partition method in the IFU.

Response: The text is changed as fallows (P3 line 92):

With regards to activity planning, different simple and safe methods are recommended by manufacturers. The activity planning is performed using a monocompartmental model, targeting a liver absorbed dose of 80-150 Gy with glass microspheres and 60 Gy for 166Ho microspheres [3]. For resin microspheres, the activity planning is often performed with a semi empirical method, using a calculation based on the patient’s body surface area and the tumor burden. In alternative, Sirtex Medical recommends also the use of a more complex method named the partition model or multicompartmental model taken into account separately the absorbed doses to the non tumoral liver, to the tumor(s) and to the lungs using MAA pre-therapy imaging [14].

  1. Table 5 is difficult to understand, numbers in square brackets are not described in the table foot.

Response: Indeed, this table is not well illustrated and explained. The table was modified and additional explanations were added in the manuscript as fallows (P8 line 264):

« Table 5 estimates the confidence intervals of the real (post-therapy) tumor absorbed dose by targeting a precise MAA tumor absorbed dose (derived from data in table 3). In worse scenarios, the real tumor absorbed dose can be significantly lower than the MAA tumor absorbed dose and being ineffective. The recommended tumor absorbed doses targeted with MAA imaging are at least 205 Gy with glass microspheres and 100 Gy with resin microspheres[16,17]. These cutoffs must be considered as a minimum and be increased as much as possible (depending on the liver tolerance). “

  1. Page 9, line 263: “Using the multicompartmental model, an alternative strategy can be used: targeting the maximum tolerable absorbed dose to the healthy liver independently of a tumor threshold dose”. It seems more accurate to make this recommendation once it is known that the tumor is going to receive at least the minimum tumoricidal dose for each type of microsphere.

Response: Large data demonstrated that MAA tumor absorbed doses cutoffs were predictive of efficacy (table 4) and therefore were established as recommended threshold doses to be used with the multicompartmental model.

Nevertheless, data demonstrated also that MAA tumor doses can be similarly over or underestimated (table 3) and therefore some patients with a low (possibly ineffective) MAA tumor absorbed dose can have finally a high (effective) tumor absorbed dose. Therefore, thresholds of MAA tumor absorbed doses must be evaluated with caution and not used necessarily for the selection of patients for SIRT. The moderate accuracy of MAA tumor dosimetry allows nevertheless predicting the level of tumor absorbed dose and therefore patients with low MAA tumor absorbed dose

would have in a similar range of low tumor absorbed dose with post therapy dosimetry, staying therefore ineffective. A compromise regarding the patient selection was added in the manuscript and a paragraph was modified as fallows (P8 line 261):

“Therefore, the recommended thresholds of tumor absorbed doses targeted with MAA imaging must be considered with caution and one should always aim for the highest tumor absorbed dose taking into account the liver tolerance). The DOSISPHERE-01 trial illustrated well this concept adopting an interesting method of activity planning in the arm of patients treated with the multicompartmental model, trying to reach very high levels of tumor absorbed doses, beyond the recommended thresholds [10]. Indeed, the activity was planned to target a tumor absorbed dose over 205 Gy but if possible over 250 Gy. In this study arm, the mean tumor absorbed dose planned with MAA SPECT/CT was finally 332 Gy, largely above the recommended thresholds. Thereafter, the clinical efficacy of this method was strongly confirmed during the follow up.

Moreover, due to the low predictive value of MAA imaging for tumor absorbed dose, thresholds of MAA tumor absorbed doses cannot be used for the patient selection during the workup. Indeed, a tumor absorbed dose threshold not reached with pre-therapy MAA dosimetry can be finally reached with post therapy dosimetry. This was demonstrated by previous data analysing post-therapy 90Y tumor dosimetry, showing a clinical efficacy under these MAA recommended tumor absorbed doses thresholds [13].

Using the multicompartmental model, an alternative strategy can be used: targeting the maximum tolerable absorbed dose to the non-tumoural liver [12,34]. This method can be used in patients having a TNR significantly higher than 1.”

  1. Other minor comments are:

-page 1, line 42: Sirtex Medical Ltd. has change to Sirtex Medical GmBH

Response: Done

-The use of more commonly used nomenclatures and therefore more easily recognizable is recommended, such as TN or TNR for Tumor/Lesion (instead of TL) or CGC for Cholangiocarcinoma (instead of CK)

Response: Done

-please, change pneumonia by pneumonitis (page 4, line 125)

Response: Done

Reviewer 2 Report

Timely topic regarding Tc99 MAA as a surrogate for y-90 radioembolization, which is especially important in this era of personalized dosimetry.  

It has been known for years that Tc99 MAA is an imperfect surrogate and could lead to falsely elevated calculation of lung shunt that either could either deny a patient of necessary treatment or result in unnecessary dose reduction and poor results. The DOSISPHERE trial demonstrated that the target dose is even higher than what was originally utilized and an unnecessary dose reduction could drop the dose below the therapeutic range.  This article represents a good summary of the benefits and limitations of Tc99 MAA as a surrogate to y-90 resin and glass microspheres.

The section regarding Holmium 166 microspheres only pertains to European readers since this treatment is not yet FDA approved in the US.

Well written and organized. 

Scientifically sound review article.

Page 4 line 126 abbreviates radioembolization as RE.  All abbreviations should be spelled out the first time they are used.

Figures/Tables: adequate

References: appropriate

Author Response

Thank you very much for your review, your kind comments and for illustrating with high interests the results of the DOSISPHERE trial, especially the uneccessary tumor dose reduction using MAA dosimetry.

  • The section regarding Holmium 166 microspheres only pertains to European readers since this treatment is not yet FDA approved in the US. »

Response: This sentence was added: “Only Sir-Spheres® and Therasphere® have a FDA approval for the US market. »

  • Page 4 line 126 abbreviates radioembolization as RE.  All abbreviations should be spelled out the first time they are used.

Response: Indeed, RE was not spelled out before. To clarify more, only SIRT was used for defining radioembolization. RE was changed by SIRT page 4.

Round 2

Reviewer 1 Report

The authors have satisfactorily addressed all comments made.